# The impact of preservation solutions for static cold storage on kidney transplantation outcomes: Results of a Brazilian nationwide multicenter study

Tainá Veras de Sandes-Freitas[1,2,3]*, Lucio Requião Moura[4,5], Deise Rosa de Boni Monteiro de Carvalho[6], Valter Duro Garcia[7], Luis Gustavo Modelli de Andrade[8], Marilda Mazzali[9], Roberto Ceratti Manfro[10], Luciane Mônica Deboni[11], Elias Davi-Neto[12], Claudia Maria Costa de Oliveira[2], Frederico Castelo Branco Cavalcanti[13], Rafael Lage Madeira[14], Ronaldo de Matos Esmeraldo[3], Denise Rodrigues Simão[15], Ana Carolina Guedes Meira[16], Gustavo Fernandes Ferreira[17], Marcus Lasmar[18], Alexandre Tortoza Bignelli[19], Alvaro Pacheco-Silva[20], José Medina Pestana[4,5], Hélio Tedesco Silva[4,5], on behalf of the DGF-Brazil Study Group[¶]

1 Departamento de Medicina Clínica, Universidade Federal do Ceará, Fortaleza, Ceará, Brazil, 2 Serviço de Nefrologia e Transplante Renal, Hospital Universitário Walter Cantídio, Fortaleza, Ceará, Brazil, 3 Setor de Transplantes, Hospital Geral de Fortaleza, Fortaleza, Ceará, Brazil, 4 Hospital do Rim, Fundação Oswaldo Ramos, São Paulo, São Paulo, Brazil, 5 Nephrology Division, Universidade Federal de São Paulo, São Paulo, São Paulo, Brazil, 6 Centro Avançado de Transplante de Órgãos e Tecidos, Hospital São Francisco na Providência de Deus, Rio de Janeiro, Rio de Janeiro, Brazil, 7 Centro de Transplantes, Santa Casa de Misericórdia de Porto Alegre, Porto Alegre, Rio Grande do Sul, Brazil, 8 Departamento de Medicina Interna, Universidade Estadual Paulista, Botucatu, São Paulo, Brazil, 9 Disciplina de Nefrologia, Faculdade de Ciencias Médicas, Universidade Estadual de Campinas, Campinas, São Paulo, Brazil, 10 Serviço de Nefrologia, Unidade de Transplante Renal, Hospital de Clínicas de Porto Alegre, Porto Alegre, Rio Grande do Sul, Brazil, 11 Serviço de Transplante, Hospital Municipal São José de Joinville, Fundação Pró-Rim, Joinville, Santa Catarina, Brazil, 12 Serviço de Transplante renal, Hospital de Clínicas da Universidade de São Paulo, São Paulo, São Paulo, Brazil, 13 Unidade de Nefrologia, Real Hospital Português de Beneficência em Pernambuco, Recife, Pernambuco, Brazil, 14 Unidade de Transplante Renal, Hospital Felício Rocho, Belo Horizonte, Minas Gerais, Brazil, 15 Departamento de Transplante Renal, Hospital Santa Isabel, Blumenau, Santa Catarina, Brazil, 16 Unidade de Transplante Renal, Santa Casa Montes Claros, Montes Claros, Minas Gerais, Brazil, 17 Unidade de Transplante Renal, Santa Casa de Misericórdia de Juiz de Fora, Juiz de Fora, Minas Gerais, Brazil, 18 Serviço de Nefrologia, Hospital Universitário Ciências Médicas, Belo Horizonte, Minas Gerais, Brazil, 19 Unidade de Transplante Renal, Hospital Universitário Cajuru, Curitiba, Paraná, Brazil, 20 Hospital Israelita Albert Einstein, São Paulo, São Paulo, Brazil

¶ Membership list of the DGF-Brazil Study Group is provided in the Acknowledgments.
* taina.sandes@gmail.com

**Data Availability Statement:** All relevant data are within the paper and its Supporting information.

## Abstract

This study evaluated the current practices of selecting cold storage preservation solutions in Brazil and their impact on delayed graft function (DGF) incidence and 1-year outcomes in kidney transplant recipients. A retrospective cohort study was conducted, including 3,134 brain-dead deceased donor kidney transplants performed between 2014 and 2015 in 18 Brazilian centers. The most commonly used preservation solution was Euro-collins (EC, 55.4%), followed by Histidine-tryptophan-ketoglutarate (HTK, 30%) and Institut Georges Lopez (IGL-1, 14.6%). The incidence of DGF was 54.4%, with 11.7% of patients requiring dialysis for more than 14 days, indicating prolonged DGF. Upon adjusting for confounding variables, HTK demonstrated a significantly lower risk of DGF than EC (OR

**Funding:** The Contatti Comércio e Representações Ltda funded part of the operational costs of the study, including statistical analysis, medical writing, language revision, and article processing charges, granted to author TVSF. The funder had no role in study design, data collection and analysis, decision to publish, or preparation of the manuscript. There was no additional external funding received for this study.

**Competing interests:** The study received financial support from Contatti Comercio e Representações Ltda, The funders had no role in study design, data collection and analysis, decision to publish, or preparation of the manuscript.

$_{0.735}0.8250_{0.926}$), as did IGL-1 (OR $_{0.605}0.712_{0.837}$). Similar protective effects were observed for prolonged DGF when comparing HTK (OR $_{0.478}0.599_{0.749}$) and IGL-1 (OR $_{0.478}0.681_{0.749}$) against EC. No significant association was found between preservation solutions and 1-year death-censored graft survival. In conclusion, EC was the most frequently used cold storage perfusion solution, demonstrating a higher incidence and duration of DGF compared with HTK and IGL-1, but with no impact on 1-year graft survival.

## Introduction

Organ preservation is a critical aspect of improving transplant outcomes. There are several strategies to achieve optimal organ preservation, including reducing cold ischemia time, using a pulsatile hypothermic perfusion pump, and optimizing static cold storage. Different solutions with varying biochemical compositions, viscosities, and costs are available for cold storage preservation. The Euro-Collins (EC) solution, a modification of the pioneering Collins solution, has been available since 1977. However, the University of Wisconsin (UW) solution is now the most commonly used preservation solution in different countries and is considered the gold standard [1]. Other solutions, such as Histidine-tryptophan-ketoglutarate (HTK), Institut Georges Lopez (IGL-1), and Celsior, have also been successfully tested for renal perfusion [2]. While current evidence, based mainly on registry studies, has shown that UW and HTK are associated with a lower incidence of delayed graft function (DGF) compared with EC, no remarkable difference was demonstrated among UW, HTK, IGL-1, and Celsior [3].

The impact of solutions on outcomes other than DGF incidence, such as DGF duration, has yet to be explored. Noteworthy, the impact of preservation solutions on transplant outcomes depends on other factors that interfere with renal preservation. In Brazil, the transplant scenario is peculiar, with a high volume of annual transplants, long cold ischemia time, poor donor maintenance, and a high incidence of DGF. The organ procurement team chooses the perfusion solution without interference from the transplant teams. The UW solution is rarely used for kidney cold storage preservation, and EC still prevails in many centers due to lower costs [4].

This study aimed to describe current practices in selecting cold storage preservation solutions in Brazil and evaluate the impact of newer solutions compared with EC on post-transplant outcomes.

## Materials and methods

### Study design and patients

This study was a *post hoc* analysis of data from the DGF-Brazil Study, a multicenter cohort including 3,992 brain-dead deceased donor kidney transplants performed between 2014 and 2015 in 18 Brazilian centers. Data was initially collected from July 1, 2017, to December 31, 2019, and subsequently analyzed between January 1, 2020, and April 1, 2020. Full details of this study have been previously reported. For the main study, recipients who lost the graft for any reason or died within seven days, those who lost the graft within 30 days due to vascular thrombosis, and those who presented primary nonfunctioning grafts were also excluded [4]. For the current analysis, the data was re-accessed between December 1, 2022, and April 1, 2023, and we excluded those patients who received pulsatile hypothermic perfusion-pumped kidneys and those without information about perfusion solutions. Due to a small sample, UW-perfused kidneys were also ruled out.

### Ethical considerations

The study was reviewed and approved by the Institutional Review Board (IRB) of the Federal University of Ceará, from where the study was coordinated (approval number 2.108.244). All participating centers also obtained local IRB approval before data collection. Obtaining informed consent or its exemption occurred following the guidelines of the Declaration of Helsinki, specific national legislation, and local IRB recommendations. In all cases where informed consent was obtained, it was written. In some cases, the local IRB waived the informed consent. Researchers at participating centers collected patient data from medical records and entered it anonymously into the REDCap platform. Patient identities were rigorously protected during data processing and analysis.

### *Post hoc* analysis objective

This analysis aimed to describe the leading preservation solutions used for cold static storage in Brazil and their impact on DGF incidence and duration, length of hospital stay, and 1-year graft survival.

### Definitions

Delayed graft function was defined as dialysis requirement during the first week after kidney transplant, excluding once-off sessions on the immediate postoperative day motivated by hypervolemia or hyperkalemia [5]. The DGF duration was assessed by the time until the last dialysis session and the number of required sessions. The requirement for dialysis for more than 14 days was classified as prolonged DGF [6]. Death-censored graft loss was defined as the return to long-term dialysis therapy or retransplantation.

### Statistical analysis

Categorical variables were presented as frequency and percentage and compared using Chi-square or Fisher tests. The Kolmogorov-Smirnov test was used to verify the distribution pattern of continuous variables. As all were non-normally distributed, they were summarized as the median and interquartile range (IQR) and compared using the Kruskal-Wallis test. Death-censored graft survival was estimated using the Kaplan–Meier method and compared by the log-rank test.

Multivariate analyses to identify independent risk factors associated with DGF and prolonged DGF were performed using a Generalized Linear Mixed Model with logistic regression, adjusted for transplant center/site (random effect). For prolonged DGF, the patients who needed dialysis for 14 days or more were compared with those who needed dialysis for less than 14 days or did not present DGF. Variables with p-value <0.1 on univariate analysis were included in the multivariate models. No collinearity was detected among the variables included in the model. Multiple Imputations by Chained Equation replaced missing values, generating five imputed datasets. A significant statistical difference was assumed when the p-value was less than 0.05. Statistical analysis was performed using IBM SPSS 25.

## Results

### Population and demographics

Out of the 3,992 patients that participated in the DGF-Brazil study, 317 were excluded because their kidneys were perfused using the pulsatile hypothermic perfusion pump. Additionally, 512 patients had no information available about their preservation solution, while 29 were

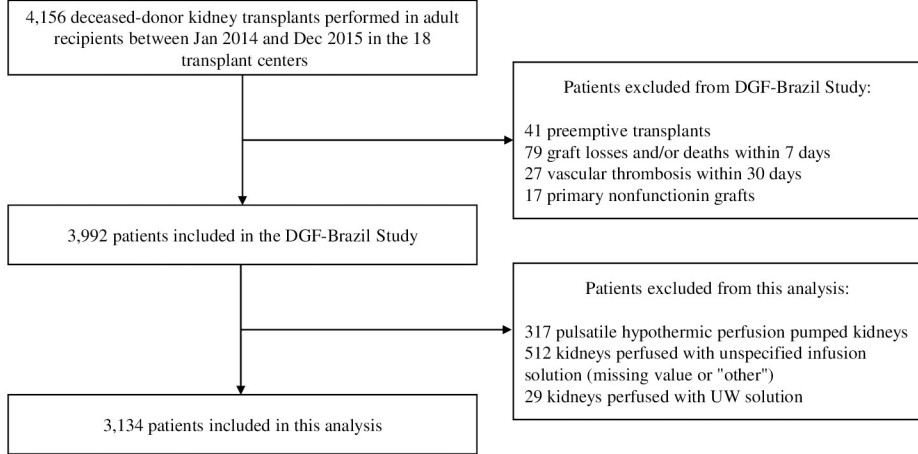

**Fig 1. Flow chart of study population.**

preserved with UW. The remaining 3,134 patients were included in this analysis (Fig 1). Table 1 summarizes the population demographics.

## Distribution of cold storage preservation solutions in Brazil

Most patients received kidneys preserved with EC (55.4%), followed by HTK (30%) and IGL-1 (14.6%). As demonstrated in Fig 2, the preservation solutions used in kidney transplants varied significantly across different Brazilian centers. The Southeast region predominantly used EC, while HTK was the prevalent solution in two of the three transplant centers in the Northeast region. In the South region, two of the five centers used EC, and IGL-1 was the leading solution in the remaining three centers.

In Table 1, we compared the demographic characteristics of patients based on the cold storage preservation solution. The groups were heterogeneous, except for the recipient age and sex, which possibly reflect different center practices and realities.

## Association between solution perfusions and transplant outcomes

The overall incidence of DGF was 54.4%, with a median duration of 8 days (IQR 4–14) and a need for 3 (IQR 2–5) dialysis sessions. Delayed graft function longer than 14 days (prolonged DGF) was observed in 11.7% of patients. The incidence of DGF in each center, based on the Brazilian Region where they are located, is demonstrated in S1 Fig.

Compared with EC and IGL-1 groups, patients in the HTK group presented a lower incidence of both DGF (HTK 44.1% vs. EC 59.5%, p<0.001; vs. IGL-1 56.6%, p<0.001) and prolonged DGF (HTK 8.2% vs. EC 13.1%, p<0.001; vs. IGL-1 13.9%, p = 0.005) and required fewer dialysis sessions (HTK 3 sessions (IQR 2–5) vs. EC 4 sessions (IQR 2–6), p = 0.002; vs. IGL-1 4 sessions (IQR 2–6), p = 0.020) (Fig 3).

Considering the time until the last dialysis session, IGL-1 was associated with a longer DGF duration [11 days (IQR 5–11)] than EC [8 days (IQR 4–14), p = 0.006] and HTK [8 days (IQR 4–13), p = 0.002]. Similar findings were observed for the length of hospital stay after transplant surgery: IGL-1 16 days (IQR 11–25) vs. EC 13 days (IQR 8–20.3), p<0.001; vs. HTK 13 days (IQR 8–21), p<0.001.

Compared with EC [47.2 mL/min/1.73m$^2$ (IQR 30.3–64.8)], both HTK [52.9 mL/min/1.73m$^2$ (IQR 36.3–73,2) p<0.001] and IGL-1 [55.4 mL/min/1.73m$^2$ (IQR 38.8–72.5), p<0.001]

**Table 1. Recipient and donor demographics, comparing patients according to preservation solutions.**

| | Non-missing data | Total N = 3,134 | EC N = 1,736 | HTK N = 940 | IGL-1 N = 458 | p-value (all groups) |
|---|---|---|---|---|---|---|
| Recipient age (years old) | 3,134 | 49.3 (38.5–58.2) | 49.4 (38.5–58.0) | 48.7 (38.7–57.9) | 50.8 (38.4–60.0) | 0.181 |
| Gender—male | 3,134 | 1,969 (62.8) | 1,089 (62.7) | 599 (63.7) | 281 (61.4) | 0.685 |
| Race | 3,066 | | | | | <0.001 |
| *Caucasian* | | 1,593 (52.0) | 859 (50.7) | 418 (45.0) | 316 (71.5) | |
| *Mixed race* | | 1,011 (33.0) | 548 (32.3) | 392 (42.2) | 71 (16.1) | |
| *Afro-Brazilian* | | 431 (14.1) | 268 (15.8) | 112 (12.1) | 51 (11.5) | |
| *Asian / Indian* | | 31 (1.0) | 20 (1.2) | 7 (0.8) | 4 (1.0) | |
| Recipient BMI (Kg/m$^2$) | 2,769 | 24.5 (21.7–27.6) | 24.4 (21.6–27.4) | 24.4 (21.7–27.7) | 25.1 (22.3–28.8) | 0.015 |
| ESKD etiology | 3,134 | | | | | <0.001 |
| *Unknown* | | 866 (27.6) | 523 (30.1) | 209 (22.2) | 134 (29.3) | |
| *Hypertension* | | 627 (20.0) | 334 (19.2) | 217 (23.1) | 76 (16.6) | |
| *Diabetes* | | 568 (18.1) | 311 (17.9) | 156 (16.6) | 101 (22.1) | |
| *Chronic GN* | | 486 (15.5) | 245 (14.1) | 193 (20.5) | 48 (10.5) | |
| *PKD* | | 244 (7.8) | 127 (7.3) | 87 (9.3) | 30 (6.6) | |
| *Other* | | 343 (10.9) | 196 (11.3) | 78 (8.3) | 69 (15.1) | |
| Time on dialysis (months) | 3,133 | 36 (19–62) | 38 (21–69) | 33 (19–58) | 27 (15–54) | <0.001 |
| Retransplantation | 3,134 | 227 (7.2) | 117 (6.7) | 59 (6.3) | 51 (11.1) | 0.002 |
| Preformed DSA>1,500 MFI | 3,030 | 185 (6.1) | 71 (38.4) | 40 (4.4) | 74 (16.3) | <0.001 |
| ECD | 3,134 | 837 (26.7) | 517 (29.8) | 24.5 (21.8) | 115 (25.1) | <0.001 |
| KDPI (%) | 3,134 | 65 (46–82) | 65 (47–83) | 63 (44–80) | 65 (45–82) | 0.030 |
| Multiple organ donor | 2,378 | 2,206 (92.8) | 1,260 (90.3) | 585 (95.0) | 361 (98.4) | <0.001 |
| CIT (h) | 3,134 | 21.0 (16.7–25.0) | 22.0 (18.0–26.0) | 20.0 (16.0–24.3) | 18.8 (15.0–22.9) | <0.001 |
| rATG induction | 3,123 | 1,980 (63.2) | 1,096 (63.2) | 661 (70.3) | 223 (48.7) | <0.001 |
| CNI-free or late introduction[1] | 3,133 | 742 (23.7) | 257 (14.8) | 364 (38.7) | 121 (26.4) | <0.001 |
| *de novo* mTORi | 3,134 | 334 (10.7) | 160 (9.2) | 147 (15.6) | 27 (5.9) | <0.001 |

Abbreviations: BMI: body mass index; ESKD: end-stage kidney disease; GN: glomerulonephritis; PKD: polycystic kidney disease; PRA: panel reactive antibodies; DSA: donor-specific anti-HLA antibodies; MFI: mean intensity fluorescence; ECD: expanded criteria donor; KDPI: Kidney Donor Profile Index; EC: Euro-Collins; HTK: Histidine-tryptophan-ketoglutarate; IGL-1: Institut Georges Lopez; CIT: cold ischemia time; rATG: rabbit antithymocyte globulin; CNI: calcineurin inhibitor; mTORi: mammalian target of rapamycin inhibitor;

[1] After 48h posttransplant

All continuous variables presented non-normal distribution and were presented as the median and interquartile range

were associated with better renal function. No differences among groups were observed in 1-year death-censored graft survival (Table 2).

## Multivariable analysis for the impact of solution perfusions on DGF incidence and duration

After accounting for confounding factors, HTK and IGL-1 showed a reduced risk of DGF by 18% and 29%, respectively, as compared to EC. Both solutions were also found to be protective against prolonged DGF, with a 51% and 32% reduced risk of needing dialysis for more than 14 days, respectively. Complete univariate and multivariate analyses are shown in Tables 3 and 4.

## Discussion

This large Brazilian nationwide multicenter study found that the most commonly used solution to preserve kidneys was EC, followed by HTK and IGL-1. The study also found that the

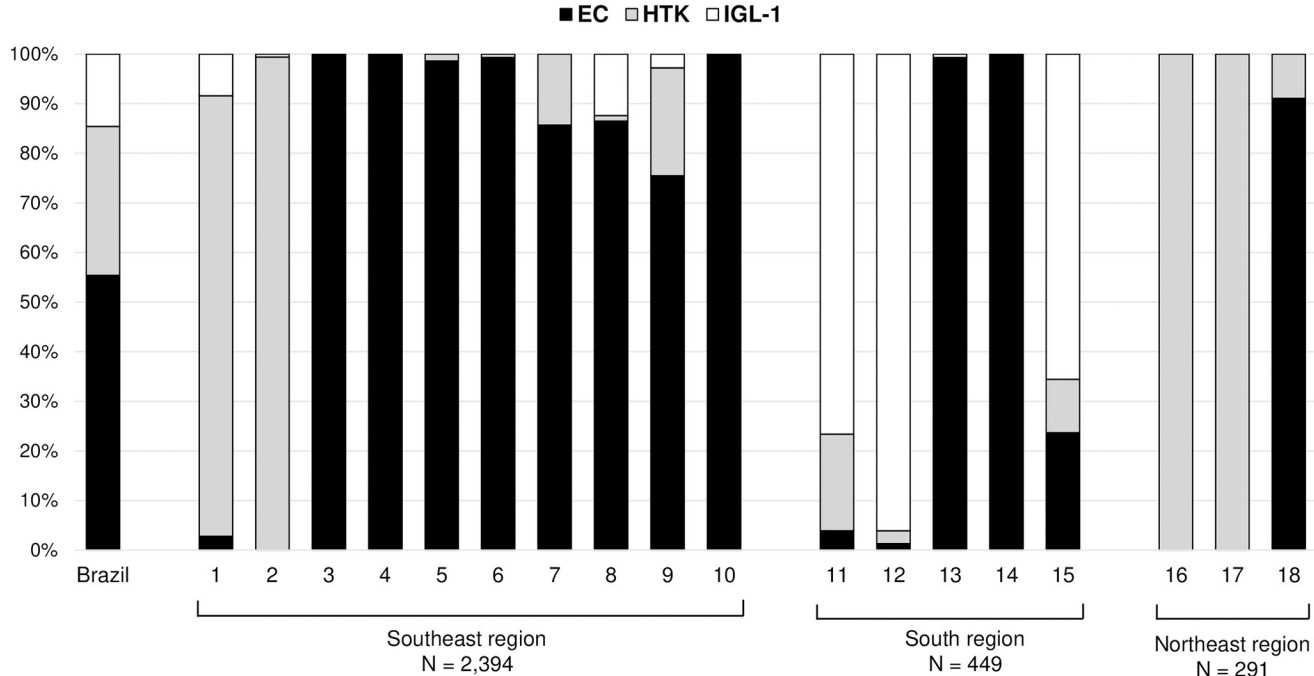

**Fig 2. Utilization trends of Euro-Collins (EC), Histidine-tryptophan-ketoglutarate (HTK), and Institut Georges Lopez (IGL-1) for kidney preservation in Brazilian transplant centers.**

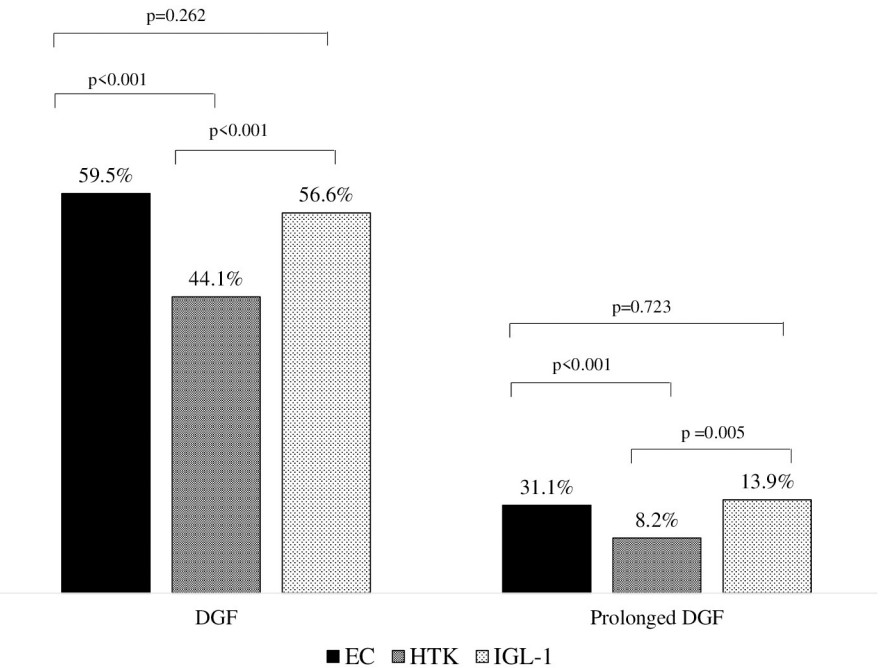

**Fig 3. Incidence of delayed graft function (DGF) and prolonged DGF in the preservation solution groups.**

**Table 2. Clinical outcomes according to the perfusion solution.**

| | Non-missing data | Total | EC | HTK | IGL-1 | P value All groups | P value EC vs. HTK | P value EC vs. IGL-1 | P value HTK vs. IGL-1 |
|---|---|---|---|---|---|---|---|---|---|
| **DGF (%)** | 3,134 | 1,707 (54.5) | 1,033 (59.5) | 415 (44.1) | 259 (56.6) | <0.001 | <0.001 | 0.262 | <0.001 |
| **Time on DGF (days)** | 1,487 | 8 (4–14) | 8 (4–14) | 8 (4–13) | 11 (5–21) | 0.005 | 0.187 | 0.006 | 0.002 |
| **Prolonged DGF#** | 2,914 | 340 (11.7) | 219 (13.1) | 75 (8.2) | 46 (13.9) | <0.001 | <0.001 | 0.723 | 0.005 |
| **No. of dialysis sessions** | 1,457 | 3 (2–5) | 4 (2–6) | 3 (2–5) | 4 (2–6) | 0.005 | 0.002 | 0.603 | 0.020 |
| **Hospitalization length (days)** | 2,768 | 13 (8.5–21) | 13 (8–20.3) | 13 (8–21) | 16 (11–25) | <0.001 | 0.992 | <0.001 | <0.001 |
| **eGFR at 12 months (mL/min/ 1.73m²)** | 3,093 | 50.7 (33.4–68.6) | 47.2 (30.3–64.8) | 52.9 (36.3–73.2) | 55.4 (38.8–72.5) | <0.001 | <0.001 | <0.001 | 0.622 |
| **1-year DCGS (%)** | 3,134 | 96.4 | 96.4 | 96.4 | 96.6 | 0.974 | 0.927 | 0.857 | 0.818 |

Abbreviations: EC: Euro-Collins; HTK: Histidine-tryptophan-ketoglutarate; IGL-1: Institut Georges Lopez; DGF: delayed graft function; eGFR: estimated glomerular filtration rate; DCGF: death censored graft survival

#Prolonged DGF: Absence of DGF or DGF duration longer than 14 days

preservation solution used for cold storage impacted the incidence and duration of DGF, favoring HTK and IGL-1 over EC-stored kidneys.

When blood flow to the graft is interrupted through vascular clamping, the subsequent cold ischemia period can lead to several adverse effects, such as depletion of adenosine triphosphate, lactic acid accumulation, and ion imbalance. These effects can cause loss of cellular

**Table 3. Risk factors for DGF.**

| | DGF (yes/no) | | | |
|---|---|---|---|---|
| | Univariate analysis | | Multivariate analysis | |
| | OR (IC 95%) | p-value | OR (IC 95%) | p-value |
| Recipient age (years-old) | 1.010 (0.008–1.012) | <0.001 | 0.979 (0.974–0.983) | <0.001 |
| Recipient gender—male | 1.264 (1.191–1341) | <0.001 | 1.243 (1.164–1.328) | <0.001 |
| Recipient BMI (Kg/m²) | 1.039 (1.032–1.046) | <0.001 | 1.047 (1.039–1.055) | <0.001 |
| Recipient race—caucasian | 1.040 (0.981–1.102) | 0.186 | NA | NA |
| Diabetic ESKR | 1.285 (1.192–1.386) | <0.001 | 1.239 (1.138–1.348) | <0.001 |
| Time on dialysis (months) | 1.005 (1.004–1.006) | <0.001 | 1.006 (1.005–1.007) | <0.001 |
| Retransplantation | 1.296 (1.158–1.451) | 0.001 | 1.246 (1.096–1.416) | <0.001 |
| DSA | 1.352 (1.196–1.529) | <0.001 | 1.256 (1.089–1.449) | 0.001 |
| Multiple organ donor | 1.004 (0.896–1.125) | 0.945 | NA | NA |
| KDPI (%) | 1.010 (1.009–1.011) | <0.001 | 1.016 (1.014–1.019) | <0.001 |
| CIT (h) | 1.049 (1.044–1.054) | <0.001 | 1.041 (1.035–1.047) | <0.001 |
| Perfusion solution | | | | |
| *EC* | REF | | REF | |
| *HTK* | 0.538 (0.504–0.574) | <0.001 | 0.825 (0.735–0.926) | 0.001 |
| *IGL-1* | 0.886 (0.814–0.964) | 0.005 | 0.712 (0.605–0.837) | <0.001 |
| rATG induction | 1.156 (1.089–1.227) | <0.001 | 0.925 (0.843–1.014) | 0.094 |
| CNI-free or late introduction | 0.842 (0.788–0.901) | <0.001 | 1.084 (0.981–1.199) | 0.115 |
| *de novo* mTORi | 0.923 (0.841–1.013) | 0.093 | 1.95 (0.973–1.232) | 0.134 |

Abbreviations: DGF: delayed graft function; ESKD: end-stage kidney disease; DSA: donor-specific anti-HLA antibodies; MFI: mean intensity fluorescence; CIT: cold ischemia time; EC: Euro-Collins; HTK: Histidine-tryptophan-ketoglutarate; IGL-1: Institut Georges Lopez; rATG: rabbit antithymocye globulin; CNI: calcineurin-inhibitor; mTORi: mammalian target of rapamycin inhibitor; OR: odds ratio; IRR: incidence rate ratio; CI: confidence interval; REF: reference.

**Table 4. Risk factors for prolonged DGF.**

| | Prolonged DGF (yes/no)[#] | | | |
|---|---|---|---|---|
| | Univariate analysis | | Multivariate analysis | |
| | OR (IC 95%) | p-value | OR (IC 95%) | p-value |
| Recipient age (years-old) | 1.003 (0.999–1.006) | 0.173 | NA | NA |
| Recipient gender—male | **1.279 (1.159–1.411)** | **<0.001** | **1.201 (1.079–1.336)** | **0.001** |
| Recipient BMI (Kg/m²) | **1.033 (1.022–1.044)** | **<0.001** | **1.022 (1.010–1.034)** | **<0.001** |
| Recipient race—caucasian | 1.058 (0.963–1.161) | 0.240 | NA | NA |
| Diabetic ESKR | **1.137 (1.011–1.278)** | **0.032** | **1.159 (1.019–1.320)** | **0.025** |
| Time on dialysis (months) | **1.003 (1.002–1.004)** | **<0.001** | **1.004 (1.003–1.005)** | **<0.001** |
| Retransplantation | **1.474 (1.251–1737)** | **<0.001** | **1.827 (1.515–2.204)** | **<0.001** |
| DSA | **1.356 (1.127–1.633)** | **0.001** | **1.410 (1.139–1.746)** | **0.002** |
| Multiple organ donor | 1.037 (0.866–1.243) | 0.694 | NA | NA |
| KDPI (%) | **1.002 (1.000–1.004)** | **0.093** | **1.005 (1.003–1.007)** | **<0.001** |
| CIT (h) | **1.017 (1.010–1.024)** | **<0.001** | **1.041 (1.031–1.050)** | **<0.001** |
| Perfusion solution | | | | |
| *EC* | **REF** | | REF | |
| *HTK* | **0.555 (0.473–0.650)** | **<0.001** | **0.599 (0.478–0.749)** | **<0.001** |
| *IGL-1* | 0.935 (0.813–1.075) | 0.347 | **0.681 (0.478–0.749)** | **0.015** |
| rATG induction | **0.840 (0.763–0.924)** | **<0.001** | 0.990(0.851–1.152) | 0.897 |
| CNI-free or late introduction | **1.167 (1.052–1.294)** | **0.003** | 1.052 (0.901-1.229) | 0.519 |
| *de novo* mTORi | **1.633 (1.436–1.857)** | **<0.001** | **2.224 (1.880–2.630)** | **<0.001** |

Abbreviations: DGF: delayed graft function; ESKD: end-stage kidney disease; DSA: donor-specific anti-HLA antibodies; MFI: mean intensity fluorescence; CIT: cold ischemia time; EC: Euro-Collins; HTK: Histidine-tryptophan-ketoglutarate; IGL-1: Institut Georges Lopez; rATG: rabbit antithymocye globulin; CNI: calcineurin-inhibitor; mTORi: mammalian target of rapamycin inhibitor; OR: odds ratio; IRR: incidence rate ratio; CI: confidence interval; REF: reference.

[#]Prolonged DGF: Absence of DGF or DGF duration longer than 14 days.

integrity, edema, and cell death. To avoid such damage, preservation solutions during organ flushing and cold storage are necessary. These solutions help to maintain cell membrane waterproofing, control electrolyte balance, and pH, and reduce the formation of oxygen-free radicals [7].

Different solutions are used for static cold storage, each with a unique biochemical composition. The EC solution, the most commonly used solution for kidney preservation in Brazil, is a phosphate-based solution with low sodium and high potassium concentration, mimicking the intracellular environment but with a high glucose concentration, high osmolarity, and low viscosity. A high potassium concentration may increase vascular resistance and hamper organ perfusion. On the other hand, the HTK solution is a low-viscosity solution that mimics the intracellular environment with low sodium but maintains low potassium concentrations. It contains histidine as a buffer, tryptophan as a free radical scavenger and membrane stabilizer, and ketoglutarate as an energy substrate. Low-viscosity solutions offer a rapid flow rate, quicker cooling of organs, and a low risk of red blood cell aggregation and vascular thrombosis. However, a larger volume of solution is required to flush out organs [8, 9].

In this cohort, HTK was superior to EC in preventing DGF, which aligns with previously published studies [3, 10]. Data with prolonged DGF were similar to DGF, indicating a potentially more significant protective effect.

The IGL-1 solution was also superior to EC in preventing DGF and prolonged DGF. To our knowledge, no previous studies compared the outcomes of kidneys perfused with IGL-1

versus EC. However, plenty of evidence indicates the superiority of the gold standard UW versus EC [3, 11]. The IGL-1 solution has many similarities to the UW solution, including containing colloids and being a viscous solution. While the colloid used and the sodium/potassium concentration differ between the two solutions (hydroxyethyl starch for UW and Poly-ethylene glycol for IGL-1; intracellular sodium/potassium concentration pattern for UW and extracellular pattern for IGL-1), the composition is otherwise similar [8].

In this study, we did not directly compare IGL-1 and HTK. Data available to date do not provide robust evidence about the superiority of either. Available studies based on retrospective analyses have conflicting results [12, 13].

Multivariate analysis to identify the impact of the perfusion solutions on the length of hospital stay was not performed because we understood that DGF directly influenced this variable. There was no association between the cold storage preservation solutions and 1-year survival. However, previous studies with longer follow-ups have demonstrated the impact of different solutions on graft survival [1]. Furthermore, in this study, HTK and IGL-1 were associated with better renal function at 12 months, a known surrogate marker for long-term renal allograft survival [14].

Previous studies have yet to thoroughly explore the significance of the time spent on dialysis after transplantation, a crucial surrogate outcome to evaluate DGF. The need for dialysis in the first week after transplantation varies greatly among different centers and is significantly influenced by local clinical protocols and logistical constraints. Therefore, requiring dialysis for more than two weeks is a more robust indicator of a severe ischemia-reperfusion injury that may have long-term consequences [6]. The high percentage of prolonged DGF in the Brazilian cohorts is noteworthy, suggesting that this phenomenon is not a consequence of more liberal dialysis practices but of insults that result in ischemia-reperfusion injuries, such as prolonged cold ischemia time and poor donor maintenance, negatively impacting transplant outcomes [4, 6, 15].

This study has some limitations, which should be pointed out a) in the primary database, transplants with primary nonfunction and early thrombosis were previously ruled out, precluding the analysis of the impact of preservation solutions on these outcomes. The decision to exclude these conditions was based on the understanding that they might indicate injuries unrelated to ischemia-reperfusion injury. Nonetheless, the number of patients excluded was minimal (n = 17) without impacting the interpretation of the results. b) The analysis focused solely on 1-year outcomes. It is well-known that DGF and prolonged DGF can significantly affect long-term graft survival [4]; c) this is a historical record based on transplants performed in the years 2014 and 2015. However, as far as we know, in recent years, no significant changes with potential impact on these results have occurred in Brazilian transplant programs; d) Our study focused on evaluating the clinical impact of perfusion solutions on graft function without the intention of providing mechanistic insights into why different preservation solutions have a distinct impact on graft health; d) finally, the use of EC solution is currently restricted to select countries, thereby limiting the generalizability of our study findings to nations employing more modern preservation solutions like UW. However, among the top four countries globally regarding transplant volume, three are classified as middle-income (China, India, and Brazil), where economic factors significantly influence healthcare practices. About 6,000 kidney transplants are performed in Brazil annually, with over 90% funded by public resources, and transplant programs have been facing underfunding in recent years [16]. This study provides real-world evidence to support pharmacoeconomic analyses, aiding decision-making processes regarding selecting the most suitable perfusion solution in Brazil and other resource-constrained countries.

In conclusion, EC was the most prevalent solution used for cold storage kidney preservation. This solution was associated with higher DGF incidence and duration compared to HTK and IGL-1, without impacting on 1-year allograft survival.

## Supporting information

**S1 Fig. Incidence of delayed graft function in Brazilian transplant centers.**
(TIF)

**S1 File.**
(XLSX)

## Acknowledgments

The authors thank the support of the Associação Brasileira de Transplantes de Órgãos (ABTO), and the DGF-Brazil Study Group, which includes the following individuals:

Alberto Rafael Baleeiro Silva (Unidade de Transplante Renal, Santa Casa Montes Claros, Montes Claros, MG, Brazil), Alessandra Rosa Vicari (Serviço de Nefrologia, Unidade de Transplante Renal, Hospital de Clínicas de Porto Alegre, Porto Alegre, RS, Brazil), Alexandre Tortoza Bignelli (Unidade de Transplante Renal, Hospital Universitário Cajuru, Curitiba, PR, Brazil), Alvaro Pacheco-Silva (Hospital Israelita Albert Einstein, São Paulo, SP, Brazil), Ana Carolina Guedes Meira (Unidade de Transplante Renal, Santa Casa Montes Claros, Montes Claros, MG, Brazil), Ana Cristina Carvalho de Matos (Hospital Israelita Albert Einstein, São Paulo, SP, Brazil), Carlucci Guarberto Ventura (Serviço de Transplante renal, Hospital de Clínicas da Universidade de São Paulo, São Paulo, SP, Brazil), Claudia Fagundes Centro Avançado de Transplante de Órgãos e Tecidos, Hospital São Francisco na Providência de Deus, Rio de Janeiro, RJ, Brazil, Claudia Maria Costa de Oliveira (Serviço de Nefrologia e Transplante Renal, Hospital Universitário Walter Cantídio, Fortaleza, CE, Brazil), Claudia Rosso Felipe (Hospital do Rim, Fundação Oswaldo Ramos, São Paulo, SP, Brazil), Deise Rosa de Boni Monteiro de Carvalho Centro Avançado de Transplante de Órgãos e Tecidos, Hospital São Francisco na Providência de Deus, Rio de Janeiro, RJ, Brazil, Denise Rodrigues Simão (Departamento de Transplante Renal, Hospital Santa Isabel, Blumenau, SC, Brazil), Eduardo José Tonato (Hospital Israelita Albert Einstein, São Paulo, SP, Brazil), Elias David-Neto (Serviço de Transplante renal, Hospital de Clínicas da Universidade de São Paulo, São Paulo, SP, Brazil), Euler Pace Lasmar (Serviço de Nefrologia, Hospital Universitário Ciências Médicas, Belo Horizonte, MG, Brazil), Fabiana Agena (Serviço de Transplante renal, Hospital de Clínicas da Universidade de São Paulo, São Paulo, SP, Brazil), Fernanda Quadros Mendonça Marques (Unidade de Transplante Renal, Santa Casa Montes Claros, Montes Claros, MG, Brazil), Frederico Castelo Branco Cavalcanti (Unidade de Nefrologia, Real Hospital Português de Beneficência em Pernambuco, Recife, PE, Brazil), Geraldo Sérgio Gonçalves Meira (Unidade de Transplante Renal, Santa Casa Montes Claros, Montes Claros, MG, Brazil), Gustavo Ferreira (Unidade de Transplante Renal, Santa Casa de Misericórdia de Juiz de Fora, Juiz de Fora, MG, Brazil), Gustavo Rocha de Oliveira (Unidade de Transplante Renal, Hospital Felício Rocho, Belo Horizonte, MG, Brazil), Hélio Tedesco-Silva (Hospital do Rim, Fundação Oswaldo Ramos / Nephrology Division, Universidade Federal de São Paulo, São Paulo, SP, Brazil), Hong Si Nga (Departamento de Medicina Interna, Universidade Estadual Paulista, Botucatu, SP, Brazil), Humberto Rebello Narciso (Departamento de Transplante Renal, Hospital Santa Isabel, Blumenau, SC, Brazil), Ivailda Barbosa Fonseca (Unidade de Nefrologia, Real Hospital Português de Beneficência em Pernambuco, Recife, PE, Brazil), José Osmar Medina Pestana (Hospital do Rim, Fundação Oswaldo Ramos / Nephrology Division, Universidade Federal de

São Paulo, São Paulo, SP, Brazil), Juliana Bastos (Unidade de Transplante Renal, Santa Casa de Misericórdia de Juiz de Fora, Juiz de Fora, MG, Brazil), Larissa Guedes da Fonte Andrade (Unidade de Nefrologia, Real Hospital Português de Beneficência em Pernambuco, Recife, PE, Brazil), Lúcio Roberto Requião Moura (Hospital do Rim, Fundação Oswaldo Ramos / Nephrology Division, Universidade Federal de São Paulo, São Paulo, SP, Brazil), Luis Gustavo Modelli de Andrade (Departamento de Medicina Interna, Universidade Estadual Paulista, Botucatu, SP, Brazil), Luís Gustavo Trindade (Serviço de Nefrologia, Hospital Universitário Ciências Médicas, Belo Horizonte, MG, Brazil), Luciane Mônica Deboni (Serviço de Transplante, Hospital Municipal São José de Joinville, Fundação Pró-Rim, Joinville, SC, Brazil), Marcos Alexandre Vieira (Serviço de Transplante, Hospital Municipal São José de Joinville, Fundação Pró-Rim, Joinville, SC, Brazil), Marcus Lasmar (Serviço de Nefrologia, Hospital Universitário Ciências Médicas, Belo Horizonte, MG, Brazil), Marcos Vinicius de Sousa (Disciplina de Nefrologia, Faculdade de Ciencias Médicas, Universidade Estadual de Campinas, Campinas, SP, Brazil), Mariana Ferneda Puerari (Unidade de Transplante Renal, Hospital Universitário Cajuru, Curitiba, PR, Brazil), Mariana Moraes Contti (Departamento de Medicina Interna, Universidade Estadual Paulista, Botucatu, SP, Brazil), Marilda Mazzali (Disciplina de Nefrologia, Faculdade de Ciencias Médicas, Universidade Estadual de Campinas, Campinas, SP, Brazil), Paula Frassinetti Castelo Branco Camurça Fernandes (Serviço de Nefrologia e Transplante Renal, Hospital Universitário Walter Cantídio), Rafael Lage Madeira (Unidade de Transplante Renal, Hospital Felício Rocho, Belo Horizonte, MG, Brazil), Roberto Ceratti Manfro (Serviço de Nefrologia, Unidade de Transplante Renal, Hospital de Clínicas de Porto Alegre, Porto Alegre, RS, Brazil), Ronaldo de Matos Esmeraldo (Setor de Transplantes, Hospital Geral de Fortaleza, Fortaleza, CE, Brazil), Roger Kist (Centro de Transplantes, Santa Casa de Misericórdia de Porto Alegre, Porto Alegre, RS, Brazil), Sandra Simone Villaça (Unidade de Transplante Renal, Hospital Felício Rocho, Belo Horizonte, MG, Brazil), Silvana Daher da Costa (Serviço de Nefrologia e Transplante Renal, Hospital Universitário Walter Cantídio / Setor de Transplantes, Hospital Geral de Fortaleza, Fortaleza, CE, Brazil), Silvia Regina Hokazono (Unidade de Transplante Renal, Hospital Universitário Cajuru, Curitiba, PR, Brazil), Sônia Leite da Silva (Serviço de Nefrologia e Transplante Renal, Hospital Universitário Walter Cantídio), Tainá Veras de Sandes-Freitas (Serviço de Nefrologia e Transplante Renal, Hospital Universitário Walter Cantídio / Setor de Transplantes, Hospital Geral de Fortaleza / Departamento de Medicina Clínica, Universidade Federal do Ceará, Fortaleza, CE, Brazil), Tereza de Azevedo Matuck (Centro Avançado de Transplante de Órgãos e Tecidos, Hospital São Francisco na Providência de Deus, Rio de Janeiro, RJ, Brazil), Valter Duro Garcia (Centro de Transplantes, Santa Casa de Misericórdia de Porto Alegre, Porto Alegre, RS, Brazil).

Lead author of DGF-Brazil Study Group: Tainá Veras de Sandes Freitas– taina.sandes@gmail.com

## Author Contributions

**Conceptualization:** Tainá Veras de Sandes-Freitas, Hélio Tedesco Silva.

**Data curation:** Tainá Veras de Sandes-Freitas.

**Formal analysis:** Tainá Veras de Sandes-Freitas.

**Funding acquisition:** Tainá Veras de Sandes-Freitas.

**Investigation:** Tainá Veras de Sandes-Freitas, Lucio Requião Moura, Luis Gustavo Modelli de Andrade, Hélio Tedesco Silva.

**Methodology:** Tainá Veras de Sandes-Freitas, Hélio Tedesco Silva.

**Project administration:** Tainá Veras de Sandes-Freitas.

**Resources:** Tainá Veras de Sandes-Freitas.

**Software:** Tainá Veras de Sandes-Freitas.

**Supervision:** Tainá Veras de Sandes-Freitas, Hélio Tedesco Silva.

**Validation:** Tainá Veras de Sandes-Freitas, Lucio Requião Moura, Hélio Tedesco Silva.

**Visualization:** Tainá Veras de Sandes-Freitas, Lucio Requião Moura, Hélio Tedesco Silva.

**Writing – original draft:** Tainá Veras de Sandes-Freitas, Lucio Requião Moura, Deise Rosa de Boni Monteiro de Carvalho, Valter Duro Garcia, Luis Gustavo Modelli de Andrade, Marilda Mazzali, Roberto Ceratti Manfro, Luciane Mônica Deboni, Elias Davi-Neto, Claudia Maria Costa de Oliveira, Frederico Castelo Branco Cavalcanti, Rafael Lage Madeira, Ronaldo de Matos Esmeraldo, Denise Rodrigues Simão, Ana Carolina Guedes Meira, Gustavo Fernandes Ferreira, Marcus Lasmar, Alexandre Tortoza Bignelli, Alvaro Pacheco-Silva, José Medina Pestana, Hélio Tedesco Silva.

**Writing – review & editing:** Tainá Veras de Sandes-Freitas, Lucio Requião Moura, Deise Rosa de Boni Monteiro de Carvalho, Valter Duro Garcia, Luis Gustavo Modelli de Andrade, Marilda Mazzali, Roberto Ceratti Manfro, Luciane Mônica Deboni, Elias Davi-Neto, Claudia Maria Costa de Oliveira, Frederico Castelo Branco Cavalcanti, Rafael Lage Madeira, Ronaldo de Matos Esmeraldo, Denise Rodrigues Simão, Ana Carolina Guedes Meira, Gustavo Fernandes Ferreira, Marcus Lasmar, Alexandre Tortoza Bignelli, Alvaro Pacheco-Silva, José Medina Pestana, Hélio Tedesco Silva.

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
