## [Decision Letter · Decision Letter 0]

3 Apr 2024

PONE-D-23-33615The impact of preservation solutions for static cold storage on kidney transplantation outcomes: results of a Brazilian nationwide multicenter studyPLOS ONE

Dear Dr. de Sandes-Freitas,

Thank you for submitting your manuscript to PLOS ONE. After careful consideration, we feel that it has merit but does not fully meet PLOS ONE’s publication criteria as it currently stands. Therefore, we invite you to submit a revised version of the manuscript that addresses the points raised during the review process.

We look forward to receiving your revised manuscript.

Kind regards,

Boris Rubinsky, Ph.D.

Academic Editor

PLOS ONE

Journal Requirements:

"This study was partially funded by Contatti Comercio e Representações Ltda."

"The study received financial support from Contatti Comercio e Representações Ltda, The funders had no role in study design, data collection and analysis, decision to publish, or preparation of the manuscript. "

4. One of the noted authors is a group or consortium [DGF Brazil Study Group]. In addition to naming the author group, please list the individual authors and affiliations within this group in the acknowledgments section of your manuscript. Please also indicate clearly a lead author for this group along with a contact email address.

5. We are unable to open your Supporting Information file renamed_cae3b.sav. Please kindly revise as necessary and re-upload.

Additional Editor Comments:

the paper should include a comparison with data from the use of UW, outside of Brazil.

Euro-Collins is not the best solution for preservation and publishing just the results with Euro-Collins may send the wrong message.

Reviewers' comments:

Reviewer's Responses to Questions

**Comments to the Author**

1. Is the manuscript technically sound, and do the data support the conclusions?

Reviewer #1: Yes

Reviewer #2: Partly

2. Has the statistical analysis been performed appropriately and rigorously? 

Reviewer #1: Yes

Reviewer #2: Yes

3. Have the authors made all data underlying the findings in their manuscript fully available?

Reviewer #1: Yes

Reviewer #2: Yes

4. Is the manuscript presented in an intelligible fashion and written in standard English?

Reviewer #1: Yes

Reviewer #2: No

5. Review Comments to the Author

Reviewer #1: Thank you for the opportunity to review your manuscript. This is a very well-written document. The introduction, methods, results and discussion are all appropriate with excellent quality of the English language, tables and figures.

A few minor comments for the authors to consider:

1- it would be great to have a flow chart illustrating the entire process of selection and exclusion of the cohort

2- please correct your definition and use: pulsatile hypothermic perfusion pump when referring to the machine used to preserve the kidneys

3- i would invite the authors to create a histogram illustrating the incidence of DGF (%) for the different preservation groups

4- similarly, it would be great for the readers to see a histogram illustrating the incidence of prolonged DGF

Reviewer #2: Manuscript: PONE-D-23-33615

The work by Veras de Sandes-Freitas et al. titled ‘The impact of preservation solutions for static cold storage on kidney transplantation outcomes: results of a Brazilian nationwide multicenter study’ investigates the kidney procurement and preservation practices throughout Brazil and the impact on delayed graft function (DGF) and one year graft survival. Their primary finding is that the most commonly used preservation solution in Brazil, Euro-collins solution, is associated with higher rates of DGF and prolonged DGF, defined as requiring dialysis for more than 14 days. No difference is seen in 1 year graft survival. Their analysis includes a multivariate analysis to confirm that the choice of preservation solution is an independent predictor of DGF. The authors clearly demonstrate these differences with their focused analyses. These results could influence the practices within Brazil and significantly impact a large number of kidney transplant recipients in that country. Below are outlined comments/concerns regarding the manuscript in its current form.

Major points:

1) The scope of the work presented in this manuscript is limited to population level analyses of kidney preservation solution. It is likely better suited for a specialty journal that focuses on transplantation as opposed to a general scientific journal.

2) The overall applicability of the findings is limited due to the different practices using preservation solutions globally. As the article mentions, University of Wisconsin (UW) solution is very widely used and is the most commonly used preservation solution in the United States. Unfortunately, because of its limited use in Brazil, it is not included in this analysis. Thus, the overall results will have limited appeal outside of Brazil

3) The analysis is limited to a retrospective post hoc study that is not structured to provide any mechanistic insight into the reason why different preservation solutions impact graft health. An understanding of these mechanisms would deepen the impact of the manuscript and make it more broadly applicable.

Minor points:

1) As pointed out in the discussion, the original study has excluded primary non-function kidneys. This issue is potentially very important and should be discussed in more depth.

2) Table 1 provides an overview of the total cohort of patients in the study, but it does not provide a comparison between the preservation groups. While this information is available in a supplemental table, it may be better situated as Table 1 in the main text in order to provide context for the remaining analyses of the article.

3) The manuscript would generally benefit from work to improve the readability.

6. PLOS authors have the option to publish the peer review history of their article (what does this mean?). If published, this will include your full peer review and any attached files.

Reviewer #1: **Yes: **Michele Molinari

Reviewer #2: No

---

## [Editor Report · Decision Letter 1]

11 Jun 2024

The impact of preservation solutions for static cold storage on kidney transplantation outcomes: results of a Brazilian nationwide multicenter study

PONE-D-23-33615R1

Dear Colleaque

We’re pleased to inform you that your manuscript has been judged scientifically suitable for publication and will be formally accepted for publication once it meets all outstanding technical requirements.

Kind regards,

Boris Rubinsky, Ph.D.

Academic Editor

PLOS ONE
---

## [Editor Report · Acceptance letter]

26 Jun 2024

PONE-D-23-33615R1 

PLOS ONE

Dear Dr. Sandes-Freitas, 

I'm pleased to inform you that your manuscript has been deemed suitable for publication in PLOS ONE. Congratulations! Your manuscript is now being handed over to our production team.

Kind regards, 

on behalf of

Professor Boris Rubinsky 

Academic Editor

PLOS ONE